# Dual-Mode Aptamer AP1-F Achieves Molecular–Morphological Precision in Cancer Diagnostics via Membrane NCL Targeting

**DOI:** 10.3390/cimb47110904

**Published:** 2025-10-30

**Authors:** Zhenglin Yang, Lingwei Wang, Chaoda Xiao, Xiangchun Shen

**Affiliations:** 1The High Efficacy Application of Natural Medicinal Resources Engineering Center of Guizhou Province, The High Educational Key Laboratory of Guizhou Province for Natural Medicinal Pharmacology and Drug Ability, School of Pharmaceutical Sciences, Guizhou Medical University, Guiyang 561113, China; crazyyang168@163.com (Z.Y.); w18984136703@163.com (L.W.); 2The State Key Laboratory of Functions and Applications of Medicinal Plants, Guizhou Medical University, Guiyang 561113, China

**Keywords:** G-quadruplex, cancer detection, pathological diagnosis, nucleolin, membrane probe

## Abstract

Nucleic acid aptamers leverage defined tertiary structures for precise molecular recognition, positioning them as transformative biomedical tools. We engineered AP1-F, a G-quadruplex (G4)-structured aptamer that selectively binds membrane-anchored nucleolin (NCL) non-permeabilizing, overcoming a key limitation of conventional probes. Microscale thermophoresis confirmed nanomolar affinity to NCL. By means of rigorous optimization, AP1-F attained a greater than ten-fold fluorescence signal ratio between malignant and normal cells in co-cultures, exceeding the extensively researched AS1411. Dual-channel flow cytometry demonstrated over 98.78% specificity at single-cell resolution within heterogeneous cell populations, owing to AP1-F’s unique membrane localization—unlike AS1411’s intracellular uptake, which elicited erroneous signals from cytoplasmic NCL. Competitive binding experiments and Laser Confocal Imaging confirmed that AP1-F specifically identifies cancer cells by binding to the NCL recognition site on the membrane. In pathological sections, AP1-F exhibited a 40.5-fold fluorescence intensity ratio between tumor and normal tissue, facilitating accurate tissue-level differentiation. Significantly, it delineated molecular subtypes by associating membrane NCL patterns with morphometric analysis: luminal-like MCF-7 displayed consistent staining in cohesive clusters, whereas basal-like MDA-MB-468 revealed sporadic NCL with irregular outlines—characteristics imperceptible to intracellular-targeted antibodies, thus offering subtype-specific diagnostic insights. This combination biochemical–morphological approach accomplished subtype differentiation with a single-step, non-permeabilized process that maintained lower cytotoxicity and tissue integrity. AP1-F enhances diagnostic accuracy by utilizing spatial confinement to eradicate intracellular interference, connecting molecular specificity to intraoperative margin evaluation or biopsy categorization.

## 1. Introduction

Early cancer detection via rapid biomarker quantification holds transformative potential for improving survival rates, intraoperative margin assessment, and biopsy stratification. Yet persistent diagnostic gaps remain—approximately 50% of malignancies evade detection until advanced stages [1,2,3]. Conventional multi-antibody procedures, such as ER/PR/HER2 panels, are constrained by three fundamental limitations: time-consuming sequential processing, significant analytical variability (exceeding 30% inter-batch coefficient variation), and structural damage due to membrane permeabilization [4]. These constraints are magnified in triple-negative breast cancer (TNBC), where biomarker absence fundamentally restricts immunohistochemical subtyping accuracy [5]. Membrane-anchored NCL emerges as a strategic diagnostic target, demonstrating dual clinical utility through surface overexpression on >85% malignancies [6,7,8,9,10,11] and spatial accessibility for molecular probes [12]. Despite antibodies’ prevailing pathological uses, their structural perturbation artifacts and incompatibility with real-time monitoring underscore the urgency of developing operationally streamlined alternatives.

Aptamers, also known as “chemical antibodies,” are DNA or RNA oligonucleotides with specific aliasing ability that couple to form unique secondary and tertiary structures [13,14]. Their numerous advantages of small size, high specificity, low immunogenicity, and ease of chemical modification make aptamers excellent recognition motifs for cancer biomarker discovery and detection [3,15,16,17,18,19,20,21,22,23].

G-quadruplexes (G4s) are present in guanine-rich DNA and RNA sequences and are formed by the π-π stacking of two or more planar Hoogsteen hydrogen-bonded G-tetrads, which are further stabilized by monovalent cations [24]. Compared to unstructured sequences, G4-forming aptamers exhibit better thermodynamic and chemical stability and are resistant to a wide range of serum nucleases [25,26]. Notably, G4 structured aptamers exhibit intrinsic NCL-binding capability [27]. Thus, there is increasing interest in developing NCL-specific aptamers that can form G4s.

Herein, we report a fluorescent G4 aptamer probe (AP1-F) designed for specific quantification of membrane NCL on live cancer cells. Unlike conventional G4 aptamers such as AS1411, which exhibit compromised specificity due to cellular internalization, AP1-F achieves spatially confined binding to cancer cell surface NCL, effectively circumventing interference from intracellular NCL pools. Through systematic optimization via four-factor, three-level orthogonal experiments (temperature, washing cycles, centrifugation parameters, and centrifugation duration), AP1-F demonstrated more than 10-fold fluorescence discrimination between malignant breast cancer cells (MCF-7, MDA-MB-468) and normal cells in co-culture systems, compared to less than 3-fold for AS1411. Mechanistic studies employing dual-channel flow cytometry and confocal imaging revealed that AP1-F maintains >98.78% concordance with RFP-labeled cancer cells in heterogeneous populations (relative error < 1.22%), validating its single-cell-level precision. The surface-exclusive splicing strategy and non-permeabilizing properties of AP1-F facilitate high-resolution delineation of tumor membranes in pathological tissues, surpassing the performance of nuclear-localized NCL antibodies in morphologically detailed discrimination—an essential advantage for distinguishing molecular subtypes of heterogeneous malignancies. This study combines structural optimization with analytical validation to establish extracellular target locking as a revolutionary approach for cancer diagnostics, offering a solid framework for the development of cost-effective probes with improved specificity for clinical use.

## 2. Materials and Methods

### 2.1. Cell Culture and Buffer Solutions

MDA-MB-468, HL-7702 and MCF-7 cells were acquired from ATCC (MA, Rockefeller, MD, USA), whereas LX-2 and HCCC-9810 cells were from the Kunming cell bank of the Chinese Academy of Sciences. MCF-7, MDA-MB-468, HCCC-9810, and LX-2 cells were cultured in DMEM medium (Gibco, New York, NY, USA) supplemented with 10% fetal bovine serum and 1% penicillin/streptomycin; all cells were cultured at 37 °C in a humidified atmosphere within a 5% CO_2_ incubator. The animal research protocol was approved by the [Guizhou Medical University Animal Care and Welfare Committee] on 2 August 2024. Project identification code: [2400619].

### 2.2. Oligonucleotide Synthesis and Sample Preparation

The oligonucleotides were synthesized at a scale of 1 µmol using phosphoramidite solid-phase chemistry on an automated DNA/RNA synthesizer (Nihon Techno-Service, Kashima, Japan). After deprotection and detachment from the support, reverse high-performance liquid chromatography (Shimadzu Co., Suzhou, China) was utilized for purification, followed by desalting on a NAPTM-10 column (GE Healthcare, Chicago, IL, USA). Two buffer solutions were employed for aptamer preparation: AP1 to AP6 and the Mut sequence in 100 mM KCl and 10 mM K_3_PO_4_ (pH 7.0), and AS1411 in 100 mM NaCl and 10 mM Tris-HCl (pH 7.0). The samples were heated to 95 °C for 5 min, then cooled to room temperature and incubated overnight at 4 °C. Oligonucleotides are conjugated at the 5′ end with FAM moieties via an amide bond linker.

### 2.3. Microscale Thermophoresis (MST) Experiment

Recombinant human NCL protein was procured from Ori-Gene Technologies (Rockville, BD, USA). For MST analysis, the protein was serially diluted using a two-fold gradient in binding buffer, generating a 12-point concentration series ranging from 4 μM to 1.95 nM across individual microcentrifuge tubes. A constant concentration (300 nM) of FAM-labeled aptamers was introduced into each dilution, followed by 15 min equilibration at ambient temperature (25 °C). The reaction mixtures were then loaded into hydrophilic capillaries and analyzed using a Monolith NT.115 MST instrument (Nano-Temper Technologies, Munich, Germany) with 20% LED power and 40% MST power settings. Binding affinity experiments result in a dose–response curve that shows how the FNorm depends on the ligand concentration. These dose–response curves can be used to calculate the affinity (K_d_) of the binding. The K_d_ model that is described here is fitted to this dose–response data, and thereby the K_d_ can be estimated. With NANO-temper analysis software (MO. Affinity Analysis), we measure the change in fluorescence after turning on the IR laser and then normalize the measured fluorescence (Fhot) by dividing it by the initial fluorescence that has been measured before the laser was turned on (Fcold). Assuming that the initial fluorescence is constant, calculate the FNorm as follows:(1)$$F_{Norm}=\frac{F_{Hot}}{F_{cold}}=\left(1-f\left(c\right)\right)unbound+f\left(c\right)bound=unbound+\left(bound-unbound\right)f\left(c\right)=\;unbound+\left(bound-unbound\right)\frac{c_{target}+c_{ligand}+K_{d}-\sqrt{{\left(c_{target}+c_{ligand}+K_{d}\right)}^{2}-4c_{target}c_{ligand}}}{2c_{target}}$$

### 2.4. Orthogonal Experiment

The orthogonal experimental design was structured to optimize four critical parameters systematically affecting staining efficacy: temperature (4 °C, 25 °C, 37 °C), washing cycles (1×, 2×, 3×), centrifugation time (5 min, 10 min, 15 min), and centrifugation speed (1000× rpm, 2000× rpm, 3000× rpm). Each parameter was assigned three discrete levels, generating an L9(3^4^) orthogonal array through SPSSAU Online Analysis Platform. Assays were systematically performed according to the predefined combinations, with raw fluorescence intensity data uploaded to the platform for multivariate statistical analysis, including polar difference evaluation, multi-factor ANOVA (α = 0.05), and Tukey’s post-hoc tests. This integrated approach identified the optimal parameter set (37 °C, 3× washes, 5 min at 3000× rpm), which was subsequently standardized for all downstream cell staining protocols.

### 2.5. Selectivity Analysis of G4s on Different Cells

Cells in the logarithmic growth phase were harvested via trypsinization, quantified by hemocytometer, and seeded onto 6-well plates at 1 × 10^5^ cells/well for 8–12 h adherence. FAM-conjugated G4 aptamers were applied in concentration gradients (0–2 μM) and incubated at 37 °C/5% CO_2_ for 40 min. Unbound probes were removed through three PBS washes (pH 7.4), followed by gentle cell detachment using 2 mM EDTA (3 min, 25 °C). Cell suspensions were pelleted by centrifugation (3000 rpm = 1000× *g*, 5 min) and resuspended in PBS for flow cytometric analysis (BD FACSCanto™ II, Franklin Lakes, NJ, USA) with FITC excitation/emission settings (λex 488 nm/λem 530 ± 15 nm). Fluorescence intensity quantification was performed using Novo-express 1.5.6 software^®^ with unstained controls for baseline correction.

### 2.6. Competitive Combination Test

Employing a flow cytometry-based competitive binding displacement assay, as documented in previous studies, we examined whether AP1 inhibit the binding of specific antibodies to NCL similarly to the established NCL-binding aptamer (AS1411), thereby confirming their specific recognition of cancer cell binding targets [28]. MDA-MB-468 and MCF-7 cells were collected with 2 mM EDTA, then homogenized in individual tubes, and exposed to aptamers (0 µM, 2 µM, 4 µM) at 4 °C for 1 h. After washing with PBS, cells were incubated with different concentrations of anti-NCL primary antibody (Ab22758, Abcam, Boston, MA, USA) for one hour at 4 °C, followed by treatment with Alexa Fluor 488-conjugated anti-rabbit IgG secondary antibody (5 µg/mL) for AP1, AS1411, and Mut sequences. The fluorescence intensity was measured using flow cytometry and analyzed with Novo-express 1.5.6 software^®^.

### 2.7. Identification and Verification of Cancer Cells in Heterogeneous Cell Populations

Breast cancer cell lines MCF-7 and MDA-MB-468 and normal cells LX-2, all in the logarithmic growth phase, were subjected to trypsin digestion, collected, and counted separately. Subsequently, the two varieties of breast cancer cells were amalgamated with normal cells in designated proportions (0, 3.125%, 6.25%, 12.5%, 25%, 50%, 100%), maintaining a total cell count of 4 × 10^5^ per well. The mixes were subsequently grown for 8 h in an incubator at 37 °C with 5% CO_2_. The cells were incubated at 37 °C in a 5% CO_2_ atmosphere for 8 to 12 h. A serum-free preparation of 1 µM AP1-F in 500 µL per well was incubated at 37 °C for 40 min in darkness, followed by three washes with PBS. The samples were then digested and collected using 2 mM EDTA for analysis via flow cytometry in the FITC channel and evaluated with Novo-express 1.5.6 software^®^. The experiment was conducted three times.

Consistent expression of RFP red fluorescent protein was attained using lentiviral transfection of the HCCC-9810 cell line. HCCC-9810 and LX-2 cells, both in the logarithmic growth phase, underwent trypsin digestion, were collected, and were counted separately. The samples were then amalgamated in designated ratios (25%, 50%, 75%) and cultured on 6-well plates at a density of 4 × 10^5^ cells per well, followed by an incubation period of 8–12 h. The cells were collected and counted individually. AP1-F was made at a concentration of 1 µM in a serum-free medium, with 500 µL per well, incubated at 37 °C for 40 min in the absence of light, and subsequently photographed using a fluorescence microscope (Leica, Wetzlar, Germany). Cells were ultimately extracted using digestion with 2 mM EDTA, and fluorescence intensity was evaluated using flow cytometry after resuspension in PBS. In the same cell population, AP1-F was detected using the FITC channel, whereas RFP was identified through the PE channel. Relative average error:(2)$$\;\frac{\delta }{\mu }=\frac{\left|x-\right.\left.\mu \right|}{\mu }\times 100\%$$

### 2.8. Laser Confocal Imaging

Images of all cells stained with AP1-F, AS1411-F, or Mut sequence-F (1 µM) were captured using an Olympus VS200 confocal microscope (Olympus Corporation, Tokyo, Japan). The cells were injected onto six-well plates and incubated for 10 to 12 h. Subsequently, AP1-F, AS1411-F, or Mut sequence-F was added to the cells, incubated at 37 °C for 40 min in the absence of light, and then rinsed three times with PBS. The specimens were then fixed with 4% paraformaldehyde and stained with DAPI (10 µg/mL) for nuclei and DID (10 µg/mL) for the cell membrane. Confocal microscopy employed excitation wavelengths of 405 nm for DAPI, 488 nm for FAM, and 561 nm for DID to acquire the pictures. The images were obtained via confocal microscopy (Olympus, Tokyo, Japan) after applying an anti-fluorescence quenching agent to seal the slides.

### 2.9. MTT Assay

MCF-7 and LX-2 cells were digested in the logarithmic growth phase with 0.25% trypsin, then centrifuged at 800 revolutions per minute for 4 min to harvest cells. Fresh media was introduced and the cells were carefully resuspended. Cell concentration was determined, then cells were inoculated at a density of 5000 cells per well in a 96-well plate for ongoing culture for 24 h. After the cells adhered for 24 h, they were treated with gradient concentrations of AS1411 (0–64 μM) or AP1 (0–64 μM) in a 5% CO_2_, 37 °C incubator for 3 days. After adding 15 μL of MTT to each well and reacting for 4 h, the liquid was discarded while retaining the bottom crystals. Then, 150 μL of DMSO was added followed by incubation in the dark at 37 °C for 30 min. Once the crystals were completely dissolved, a microplate reader was used to measure the optical density (OD) of each well at 490 nm. Then, GraphPad Prism 9 software was used to analyze the IC50 values of AS1411 and AP1.

### 2.10. Cell Membrane NCL Double Immunofluorescence Analysis

Two types of breast cancer cells in the logarithmic growth phase were collected using trypsin digestion, and 1 × 10^5^ cells per well were allocated in 12-well plates containing coverslips and cultured adherently for 24 h. Serum-free AP1-F was generated at a concentration of 1 µM, 400 µL per well, and incubated at 37 °C for 40 min, subsequently undergoing three washes with PBS, fixation in 4% paraformaldehyde for 20 min, and blocking with 2% BSA for 1 h. The anti-NCL primary antibody (Ab22758, Abcam, Boston, MA, USA) was incubated at 4 °C for 24 h. Cy3-conjugated goat anti-rabbit IgG (H + L) secondary antibody (Beyotime, Shanghai, China, A0516), diluted 1:1000, was incubated at room temperature for one hour. Nuclei were stained with DAPI (10 µg/mL) for 10 min, followed by blocking the slices using an anti-fluorescence quencher. Laser Confocal Imaging Analysis (Olympus, Tokyo, Japan) employed excitation wavelengths of 405 nm for DAPI, 488 nm for AP1-FAM, and 561 nm for NCL-Cy3.

### 2.11. Immunofluorescence Staining of Pathological Tissue

Two distinct molecularly characterized breast cancer cell lines (MCF-7 and MDA-MB-468) were individually implanted into 5-week-old nude mice, which were subsequently housed in SPF-grade environments for 15 days to establish xenograft tumor models. Distinct frozen sections were prepared from the two xenograft tumor types. Frozen sections were thawed from −20 °C to room temperature and subsequently washed three times with PBST for three minutes each. Two percent BSA was employed for blocking for one hour at room temperature, followed by incubation with AP1-F or Mut-F at 1 µM or anti-NCL for one hour at room temperature, and subsequently treated with Cy3-conjugated goat anti-rabbit IgG (H + L) secondary antibody (Beyotime, A0516, dilution 1:1000) for one hour at room temperature. The cell nuclei were stained with DAPI solution for 10 min, after which the sections were sealed with Anti-fluorescence Quenching Sealer. Imaging analysis was conducted using a fluorescence microscope (Leica, Japan), with the purple excitation light corresponding to DAPI and the blue excitation light corresponding to AP1-F.

## 3. Results

### 3.1. Evaluating the Binding Affinity of G4s to NCL

The NCL effectively recognizes structure-specific families of nucleic acids, particularly G4s in DNA and RNA [29]. Studies demonstrate that G-end telomeric RNA sequences can generate G4 structures, with loop length serving as a critical determinant affecting G4 affinity for NCL [30]. Utilizing nuclear magnetic resonance, circular dichroism, and gel electrophoresis techniques, we discerned that the human telomeric RNA sequence r(GUUAGGGU) establishes a distinctive topological structure focused on a very stable G-quadruplex. Consequently, we engineered RNA sequences with differing loop lengths and previously established that these aptamers assume distinct G4 conformations (Appendix A) [31,32]. To investigate their potential for NCL binding in cancer cell recognition, six G4 sequences were labeled with FAM fluorescent motifs (referred to as APs-F), and an MST assay was conducted to evaluate the interaction between G4 molecules and NCL proteins, aiming to identify G4 candidates with the highest binding affinity for NCL (Figure 1). The K_d_ values for these G4s’ interactions with NCL were in the nanomolar range, with AP1-F having the strongest binding affinity to NCL proteins, as shown by its K_d_ value of 4.74 × 10^−7^ ± 4.77 × 10^−8^ M (Appendix A). These data indicate that all G4s exhibit a substantial affinity for NCL proteins at the molecular level.

### 3.2. Orthogonal Test to Optimize Dyeing Conditions

NCL is prominently expressed on the membrane surface of numerous proliferating cancer cells, but its expression is minimal on the membrane surface of normal cells [33]. This study aimed to validate whether G4 structures could selectively target cancer cells via NCL binding, thereby providing a foundation for developing tumor-specific staining agents. To identify a high-affinity molecular probe, AP1-F—a G4 analog exhibiting the highest in vitro binding affinity to NCL—was selected as the fluorescent probe. Staining parameters were systematically optimized using a three-level, four-factor orthogonal experimental design (staining temperature, number of washes, centrifugation speed, and centrifugation time; Appendix A). An orthogonal table comprising 10 groups of experimental conditions was generated via the SPSSAU platform (Appendix A), with tumor cells as the positive control and normal cells as the negative control.

Flow cytometry results (Figure 2) demonstrated that the average fluorescence intensity of tumor cells was significantly higher than that of normal cells across all experimental groups (** *p* < 0.01). Through polarity analysis and multifactorial ANOVA, this study clarified the optimal combinations of key parameters: centrifugation speed of 3000 rpm, washing cycles of 3 times, centrifugation duration of 5 min, and staining temperature of 37 °C (Appendix A). Statistical results indicated that the washing cycle was a significant influencing factor (* *p* < 0.05, Appendix A). Validation experiments based on orthogonal experimental design confirmed the reliability of this parameter combination, and its analytical results were in excellent agreement with theoretical predictions.

### 3.3. Screening for the Best G4 Aptamer

Based on the optimized staining protocol, the study further explored the difference in selectivity of cancer cells with different G4 structures. Using concentration gradient experiments, it was found that AP1-F showed dose-dependent fluorescence enhancement in tumor cells, and the signal tended to saturate when the concentration exceeded the threshold, suggesting that the target binding had reached a plateau phase (Figure 3); while normal cells only showed weak signal enhancement at high concentrations and stabilized at the background level. Notably, AP1-F showed the greatest fluorescence difference between tumor and normal cells, with a signal intensity ratio of more than 20-fold, which was significantly higher than that of AS1411-F by 2.4-fold (Appendix A). This finding suggests that the sequence specificity of G4 structures is intricately linked to their selectivity, with AP1-F potentially attaining high-precision recognition of cancer cell markers via particular spatial conformations. Orthogonal experiments developed a uniform technique for G4 probe staining, mitigating external environmental influences that could influence G4-specific detection of cancer cells. AP1-F demonstrates exceptional selectivity for cancer cells, establishing a foundation for its practical use in contexts such as tumor liquid biopsy and intraoperative navigation.

### 3.4. Confirm Specificity of Target Recognition

It is important to note that due to the absence of physiological conditions, such as tissue compression, the conformation of NCL protein at the molecular level in vitro may differ from its conformation at the cellular level [32]. Despite AP1-F demonstrating a strong binding affinity for NCL at the molecular level, its capacity to efficiently and stably adhere to the NCL target on the cell membrane necessitates more confirmation. Utilizing a flow cytometry-based competitive binding occupancy assay, we further substantiated NCL as a pivotal target for AP1 in the particular identification of cancer cells. An established NCL-binding aptamer (AS1411) was utilized as the positive control, whereas a mutant sequence unable to generate G4 structures functioned as the negative control (Mut sequence). The findings indicate that the fluorescence intensity of both cancer cell lines decreased in a dose-dependent manner with increasing AP1 concentration. At an AP1 concentration of 4 μM, fluorescence intensity decreased by approximately 50% relative to the PBS control group (*** *p* < 0.001), similar to the inhibitory effect observed in the positive control group (AS1411). This result demonstrates that AP1 efficiently inhibits future antibody binding to NCL by specifically binding to NCL. In contrast, cancer cells in the negative control group (Mutant) had no significant reduction in fluorescence intensity, indicating a lack of specific binding affinity to NCL. These further investigations further validate its potential importance as a crucial tool for recognizing NCL on cell surfaces and underscore the vital role of the G4 structure in identifying cancer cells. Overall, both MCF-7 and MDA-MB-468 cells exhibited a significant reduction in fluorescence intensity at different antibody concentrations following treatment with AP1 and AS1411, indicating that both agents effectively impeded additional antibody binding after interacting with NCL (Figure 4). This result confirms AP1’s exclusive affinity for NCL compared to other targets at the cellular level and illustrates that both breast cancer cell lines exhibit similar competitive response patterns to AP1-NCL binding.

### 3.5. Detecting Cancer Cells in Heterogeneous Cell Populations

To evaluate the efficacy of AP1-F in identifying cancer cells within heterogeneous populations, we combined the specified cancer cell lines with normal cells at varying ratios (0%, 3.125%, 6.25%, 12.5%, 25%, 50%, 100%), stained them with AP1-F and AS1411-F, and subsequently analyzed them using flow cytometry. The findings indicated that as the ratio of MCF-7 or MDA-MB-468 cells in the mixed cell population progressively rose, AP1-F initially identified a positive peak of cancer cells, which intensified with the increasing proportion of cancer cells. Conversely, AS1411-F had negligible positive cell peaks and displayed a delayed response (Figure 5a,c). When the cancer cell percentage attained 100%, the fluorescence intensity resulting from AP1-F labeling increased dramatically by 10.43-fold and 17.53-fold, respectively, in comparison to the initial 0% cancer cell proportion group (Figure 5b). Conversely, the AS1411-F-treated group exhibited a marginal increase in fluorescence intensity, with enhancements of 1.75-fold and 1.01-fold, respectively (Figure 5d). Control studies demonstrated that no positive signals were observed in the group treated with the mutant aptamer Mut-F (Appendix A). The results collectively demonstrate that AP1-F has improved cancer cell detection in complex heterogeneous cell systems compared to conventional aptamers.

### 3.6. Verify the Ability of AP1-F to Precisely Identify Cancer Cells

To further ascertain the accuracy of AP1-F in detecting cancer cells, we selected rectal cancer cells HCCC-9810, which stably express RFP fluorescent protein, as a tool cell and mixed them with normal cells lacking RFP expression at a specific ratio (25%, 50%, and 75%). Then, we labeled them with AP1-F staining and performed dual-channel flow cytometry to analyze the difference between the number of AP1-F-positive cells and the number of RFP-positive cells. The results of the flow cytometry dual-channel assay showed that the relative errors of FITC (AP1-F):PE (RFP)-labeled cancer cell counts in heterogeneous cell populations with different mixing ratios (25%, 50%, and 75%) were 1.22% ± 0.10%, 1.05% ± 0.21%, and 0.61% ± 0.38%, respectively (Figure 6a,b). Using the PE channel as a control, the accuracy of AP1-F-labeled positive cell counts in different mixing ratio gradients exceeded 98.78%, indicating that AP1-F possesses the ability to accurately detect cancer cells in complex heterogeneous cell populations. Furthermore, we observed the tool cells double-labeled with AP1-F and RFP using microscopy in the mixed cell population (Figure 6c). The results demonstrate that Ap1-F effectively labels tool cells within heterogeneous cell populations, underscoring its capacity to accurately identify cancer cells. Both AS1411-F and Ap1-F can bind to NCL; however, Ap1-F demonstrates a superior capacity to identify cancer cells within heterogeneous cell populations and differentiate them from normal cells. This distinction in cancer cell recognition merits further investigation.

### 3.7. Analysis of Different NCL-Targeting Strategies and Their Mediated Cytotoxicity

We performed a co-localization analysis utilizing the cell membrane dye DID to elucidate their distribution discrepancies at the subcellular level. Results demonstrated that the primary fluorescence of AP1-F was localized to the cell membrane of breast cancer cells MCF-7 and MDA-MB-468, but the intracellular fluorescence was nearly undetectable (Figure 7a). Nonetheless, no significant fluorescence was observed in LX-2, either intracellularly or on the surface of the cell membrane. This suggests that the high-resolution signal at the membrane is due to AP1-F binding to NCLs in the cancer cell membrane rather than permeabilization. Furthermore, fluorescence generated by Mut-F was seldom detected on the membranes or within the cells of all three species, suggesting that Mut-F lacks the capacity to identify cancer cells (Appendix A). This further highlights the essential role of the G4 structure in identifying cancer cells. In contrast to AP1-F, which is localized on the cell membrane, a significant quantity of fluorescence was detected within the AS1411-F-labeled cancer cells, indicating substantial internalization of AS1411-F. Furthermore, fluorescence was detected in normal cells, indicating that AS1411-F penetrated both cancerous and normal cells (Figure 7c). We subsequently conducted co-localization analysis of the superimposed images utilizing ImageJ (ImageJ-win 64), revealing significant overlap between the green (AP1-F) and red (DID) signals on the cell membranes of MCF-7 and MDA-MB-468, with co-localization correlation coefficients of 0.72 ± 0.04 and 0.56 ± 0.02, respectively (Figure 7b). Nevertheless, AS1411-F (green) and DID (red) infrequently co-localized on the cell membrane, exhibiting correlation values of merely 0.47 ± 0.01 and 0.31 ± 0.02 (Figure 7d). The findings demonstrated a significant disparity in the distribution of AP1-F and AS1411-F during cellular targeting, with AP1-F predominantly associating with the NCL on the cell membrane, whereas AS1411-F was extensively internalized within the cell.

Previous studies have shown that binding of AS1411-F to NCL disrupts NCL-mediated intracellular signaling and leads to cytotoxicity via endocytosis of cancer cells [34]. The MTT assay revealed a profound disparity in cytotoxic potency between compound AP1 and AS1411 (Appendix A). Notably, compound AP1 exhibited an IC50 value exceeding the predefined upper threshold of 64 µM, with cell viability remaining above 80% even at the maximum tested concentration (64 µM). In stark contrast, AS1411 demonstrated significantly higher cytotoxicity, achieving an IC50 of 30.34 µM for LX-2 (95% CI: 23.14–42.13 µM) and 29.83 µM for MCF-7 (95% CI: 25.97–34.73 µM). This >2.1-fold lower-bound potency difference (AP1 > 64 µM vs. AS1411 = 30.34 µM), coupled with the sustained high viability under AP1 treatment, unequivocally establishes that AP1 possesses a markedly reduced cytotoxic profile compared to AS1411. Collectively, these results establish a critical structure-function relationship: AS1411’s pronounced cytoplasmic internalization correlates with its potent cytotoxicity, whereas AP1’s preferential cell membrane localization mediated by its G4 structure minimizes intracellular uptake, thereby substantially reducing off-target toxicity.

### 3.8. Quantitative and Visual Analysis of NCL in Different Cell Membranes

Our findings reveal that membrane-anchored NCL governs the tumor-selective recognition of AP1. To systematically investigate the molecular determinants underlying this selectivity, we examined the spatial expression patterns of NCL in breast cancer cells. Surface-specific analyses using antibodies without membrane disruptors showed significantly elevated membrane NCL expression in MCF-7 and MDA-MB-468 cells compared to LX-2 control cells (Figure 8a). To further characterize membrane NCL distribution, we performed dual-labeling confocal microscopy with AP1-F and NCL-specific antibodies. Both MCF-7 and MDA-MB-468 cells exhibited strong co-localization signals. Distinct membrane fluorescence was observed for AP1-F (green) and NCL-Cy3 (red) (Figure 8b). Quantitative analysis of membrane fluorescence intensity showed that the fluorescence signal intensity of AP1-F-labeled cancer cells was significantly higher than that of normal cells (** *p* < 0.01, Figure 8c), confirming the ability of AP1-F to specifically target NCL on the membrane and differentiate cancer cells. Furthermore, AP1-F fluorescent labeling efficacy was comparable to that of traditional anti-NCL antibodies, demonstrating the potential application as an alternative to traditional anti-NCL antibodies.

These results establish that membrane-localized NCL serves as the critical determinant for AP1-mediated tumor recognition. The differential AP1-F binding efficacy between breast cancer subtypes directly correlates with surface NCL expression levels, confirming that extracellular membrane targeting rather than intracellular penetration enables precise cancer cell identification. This mechanism explains AP1-F’s selective recognition capabilities in MCF-7 and MDA-MB-468 systems while maintaining analytical resolution for malignant cell detection.

### 3.9. Identification and Diagnosis of Different Cancer Pathological Tissue Sections

Building upon the exceptional performance of AP1-F in cancer cell detection, this study further evaluates its diagnostic utility in the molecular subtyping of breast cancer. By establishing nude mouse xenograft tumor models derived from two breast cancer cell lines—MCF-7 (luminal type) and MDA-MB-468 (basal-like type) (Figure 9a)—we systematically analyzed the staining characteristics of AP1-F versus anti-NCL antibodies in tumor tissue cryosections. AP1-F demonstrated superior membrane localization specificity without requiring permeabilization: tumor tissues exhibited a remarkable 40.5-fold increase in fluorescence intensity compared to normal mammary tissues (Figure 9b,c). Crucially, complete signal ablation in Mut-F confirmed G4 structure dependency (Appendix A).

This study addresses a critical challenge in the molecular subtyping diagnosis of breast cancer—the precise identification of TNBC subtypes (such as MDA-MB-468) that lack specific biomarkers. Its significance lies in overcoming the limitations of current ER/PR/HER2 receptor-based diagnostic frameworks for TNBC molecular classification. Morphometric analysis demonstrated that AP1-F’s subtype-specific staining features correlate with tumor biological behavior: basal-like MDA-MB-468 exhibited membrane heterogeneity associated with invasiveness (pseudopodia formation and membrane ruffling, white/red arrows in Figure 9b), while luminal MCF-7 maintained regular membrane morphology. In contrast, conventional anti-NCL antibodies fail to deliver precise information on membrane interfaces and intricate morphological structures due to non-specific signals in the cytoplasm/nucleus resulting from penetration (Appendix A).

These findings collectively establish AP1-F’s molecular morphological discrimination capability: distinguishing luminal and basal-like subtypes through a single-step, non-permeabilizing spatial staining paradigm. This superior molecular profiling capacity surpasses conventional antibody-based detection methodologies, offering a novel approach for precision medicine. Particularly in the MDA-MB-468 model, the membrane heterogeneity identified by AP1-F demonstrates a strong correlation with aggressive phenotypes, suggesting its potential utility in histopathological assessment of tumor malignancy progression.

## 4. Discussion

Accurate detection of early-stage malignant tumors and real-time evaluation of intraoperative tumor clearance are essential for successful radical resection. The technique of cell membrane surface staining swiftly discovers and quantitatively assesses potential tumor biomarkers by the use of exogenous dyes, thereby efficiently tracking cancer cell dispersion among various populations. Addressing the difficulty of developing diagnostically useful tumor biomarkers and constructing probe molecules with targeted specificity is critically important.

Recent discoveries reveal that NCL, a tumor biomarker, is overexpressed on the membrane of neoplastic cells [35]. The overexpression of NCL on tumor cell membranes provides a beneficial “anchor site” for ligands, making it a subject of extensive research in bioassays and tumor-targeted therapeutics [36,37,38,39]. Our flow cytometry and immunofluorescence analyses demonstrated that membrane-bound NCL expression in breast cancer cells (MCF-7 and MDA-MB-468) was significantly higher than in normal cells (** *p* < 0.01, Figure 8), whereas normal cells (e.g., LX-2) showed negligible membrane NCL levels. This aligns with prior studies that have shown that NCL is aberrantly enriched on the membrane of >85% of malignant cells but is predominantly localized in the nucleus and cytoplasm of normal cells [33,35]. Such spatial heterogeneity underscores the rationale for developing membrane-restricted probes. Traditional antibody/aptamer-based detection, however, relies on cell permeabilization to access intracellular NCL. In this process, the elevated uptake diminishes the quantity of labeled ligands bound to the cell membrane. As a result, tumor cells have insufficient fluorescence intensity to be differentiated from normal cells [40,41,42,43,44,45,46,47]. In contrast, AP1-F achieves >98.78% accuracy in cancer cell identification within heterogeneous populations (Figure 6b) by exclusively targeting membrane-anchored NCL (Pearson’s co-localization coefficient: 0.72 ± 0.04, Figure 7b), thereby eliminating intracellular interference. This mechanism enables a more than 10-fold improvement in signal discrimination over the clinical benchmark AS1411 (Figure 5).

The high specificity of AP1-F stems from its unique G4 structure and non-internalizing behavior. Competitive binding assays revealed that AP1-F interacts with NCL in a concentration-dependent manner, while Mut-F, incapable of forming G4, completely lost binding activity (Figure 4). Moreover, subsequent flow cytometry and confocal testing revealed no capability for cancer cell-specific detection or visualization labeling (Appendix A). This result suggests that the G4 structure may selectively anchor to specific domains of membrane NCL. Forced internalization of AP1-F abolished its cancer-selective recognition (Appendix A), confirming that spatial restriction is pivotal for discriminating malignant from normal cells. Future studies could employ cryo-EM to resolve the AP1-F/NCL complex structure or utilize alanine-scanning mutagenesis to identify critical binding residues, further optimizing targeting efficiency. Furthermore, the identification of cancer cells within the complex circulatory system and the viability of clinical liquid samples require further experimental designs and collaborative efforts.

Current molecular typing systems based on ER/PR/HER2 receptor assays face significant limitations in the precise diagnosis of TNBC. In particular, about 15–20% of TNBC cases (e.g., the basal-like subtype represented by MDA-MB-468) lack specific surface markers [5], making it difficult to achieve accurate subtype differentiation by conventional immunohistochemistry (IHC)-based assays. This diagnostic dilemma directly affects therapeutic decisions. The membrane-restricted targeting strategy proposed in this study successfully breaks through the inherent limitations of the conventional ER/PR/HER2 detection framework by specifically identifying NCL proteins that are aberrantly overexpressed in tumor cell membranes. On the one hand, AP1-F achieved high-resolution identification of tumor molecular subtypes at the pathological tissue level (Figure 9b), and on the other hand, its one-step staining and non-permeable membrane stabilization properties greatly simplified the operation process and avoided structural damage and false-positive interference of normal cells in traditional antibody detection (Appendix A). The aptamer probe can assist tumor molecular typing by fluorescence intensity and spatial detail distribution characteristics, especially in identifying morphological details and distinguishing tumor subtypes better than traditional antibodies, providing a new tool for precise pathological diagnosis of membrane markers detected in situ. At the same time, they are formidable candidates for sensing owing to their reproducibility and scalability via solid-phase synthesis (which eliminates batch variations), reprogram ability through directed evolution and localized sequence modifications, and their ability to achieve high catalytic turnover independent of auxiliary proteins. This feature underscores the feasibility of AP1 as an alternative to antibodies. Aptamers, as an appealing category of targeting agents, can function as tiny molecules with enhanced flexibility, allowing them to approach target locations that may be unreachable by larger antibody molecules. Aptamers can be utilized in targeted therapeutics owing to their markedly high binding specificity, akin to that of antibodies, a characteristic unattainable by small-molecule medicines. Aptamers have beneficial properties that situate them in the therapeutic realm, connecting small-molecule drugs and high-molecular-weight antibodies.

This study design has certain limitations, primarily arising from the difficulties in obtaining cancer cells and normal cells from the same tissue source, along with clinical cancer tissue samples for detection in heterogeneous specimens. This study utilized xenograft models as a substitute. Ex vivo cancer cells exist in a unique internal environment compared to in vivo breast cancer cells, potentially leading to biological differences that may affect the translational significance of the findings. Furthermore, whereas competitive binding tests indicate a significant correlation between AP1-F labeling and NCL expression, further investigation is required to assess potential off-target effects and to ascertain whether AP1-F interacts with other membrane proteins [48]. Systematic design and development will yield innovative therapeutic applications aimed at optimizing pharmacokinetic properties and improving clinical efficacy, requiring enhanced collaboration.

## 5. Conclusions

We developed AP1-F, a G4 aptamer probe, that achieves precise discrimination of malignant cells via membrane-restricted targeting of overexpressed NCL. Unlike conventional antibodies requiring permeabilization, AP1-F directly binds surface NCL with single-cell accuracy (>98.78% concordance with RFP-labeled cancer cells, relative error < 1.22%, Figure 6a,b) and exhibits a more than 10-fold signal contrast in mixed populations (Figure 5). Its non-internalizing nature eliminates interference from cytoplasmic NCL and reduces off-target toxicity (cell viability > 80% at 64 μM), resolving the false-positive and structural damage issues inherent to antibody-based approaches. It is worth noting that compared with the cumbersome multi-step staining process in traditional pathology testing, the single-staining process of the AP1-F probe can simultaneously achieve efficient identification of cancer cells and their tissue in situ. The non-permeabilizing mechanism of action preserves cellular morphology and provides a new diagnostic tool for molecular subtype classification. Quantitative analysis demonstrated that the probe exhibited superior membrane clarity and discrimination capability compared to conventional chemical antibody approaches in the identification of tumor molecular subtypes (Figure 9b and Appendix A), a breakthrough performance advantage that stems from its unique surface targeting mechanism and optimized molecular design. Future efforts will focus on rational aptamer optimization and multicenter clinical validation to advance its applications in liquid biopsy and precision oncology.

## Figures and Tables

**Figure 1 cimb-47-00904-f001:**
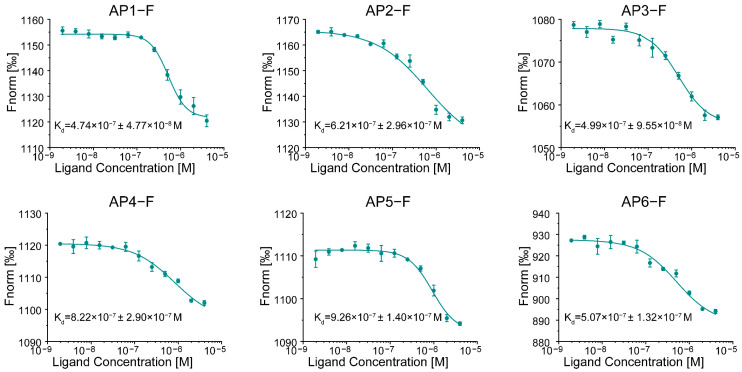
Assessment of the molecular-scale binding affinity of G4 aptamers (AP1-F to AP6-F) for NCL using MST. K_d_ values were obtained using MST binding curves. Aptamer concentration: 300 nM. NCL protein concentration range: 4.00 × 10^−6^ M to 1.95 × 10^−9^ M. Data were expressed as K_d_ ± K_d_ Confidence (n = 3).

**Figure 2 cimb-47-00904-f002:**
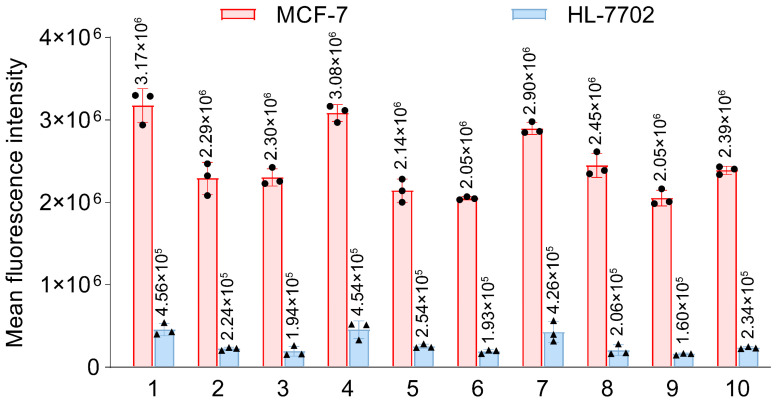
Incubated MCF-7 and HL-7702 cells with a 1 μM concentration of fluorescently tagged AP1-F, followed by treatment of the stained cells according to the specific conditions of each orthogonal experimental group (Appendix A). Examined variations in fluorescence intensity between MCF-7 and HL-7702 across each group utilizing flow cytometry. Detection channel: FITC (fixed wavelength at λex/λem = 488 nm/530 nm). Data were expressed as the mean fluorescence intensity ± SD (n = 3).

**Figure 3 cimb-47-00904-f003:**
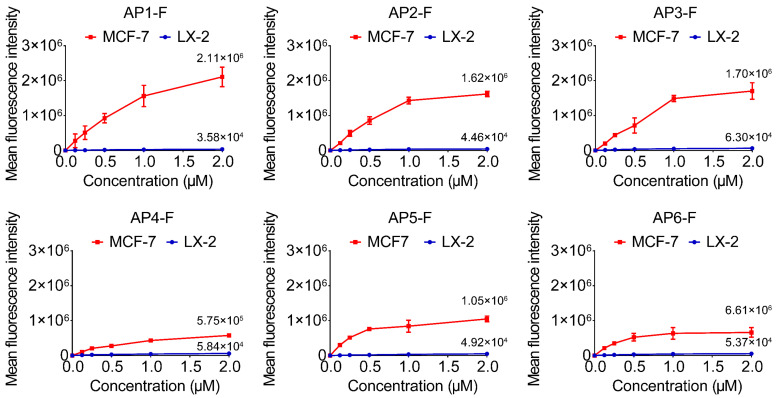
Variations in fluorescence intensity between MCF-7 cells and LX-2 normal cells at identical gradient concentrations of G4s staining circumstances (0 μM, 0.125 μM, 0.25 μM, 0.5 μM, 1.0 μM, 2.0 μM). Detection channel: FITC (fixed wavelength at λex/λem = 488 nm/530 nm). Data were expressed as the mean fluorescence intensity ± SD (n = 3).

**Figure 4 cimb-47-00904-f004:**
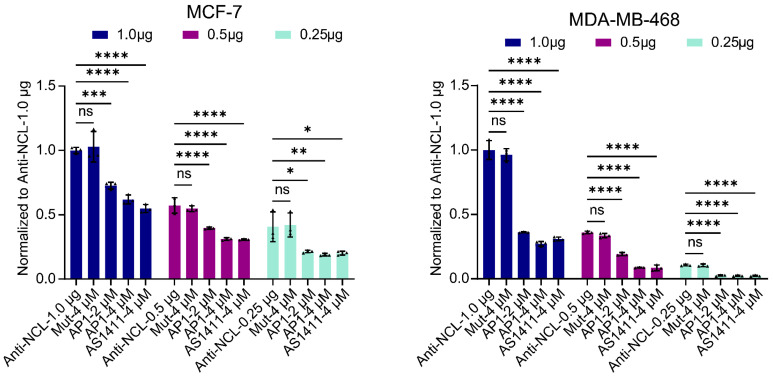
We employed the principle of competitive binding to confirm AP1-specific binding sites in two representative breast cancer cell lines, MCF-7 and MDA-MB-468. Concentrations of the primary antibody are 1 μg/100 μL, 0.5 μg/100 μL, and 0.25 μg/100 μL. The secondary antibody, Alexa Fluor 488-conjugated anti-rabbit IgG, was utilized at a concentration of 5 µg/mL. Experimental groups: Blank control group (antibodies only), negative control group (mutant: 4 μM), treatment group (AP1: 2 μM, 4 μM), and positive control group (AS1411: 4 μM). Detection channel: FITC (fixed wavelength at λex/λem = 488 nm/530 nm). The flow cytometry histogram findings were obtained at a fixed wavelength of λex/λem = 488 nm/530 nm. The data were shown as the mean fluorescence intensity ± standard error of the mean (n = 3). The statistical significance thresholds were * *p* < 0.05, ** *p* < 0.01, *** *p* < 0.001, **** *p* < 0.0001, as ascertained by a One-way ANOVA test.

**Figure 5 cimb-47-00904-f005:**
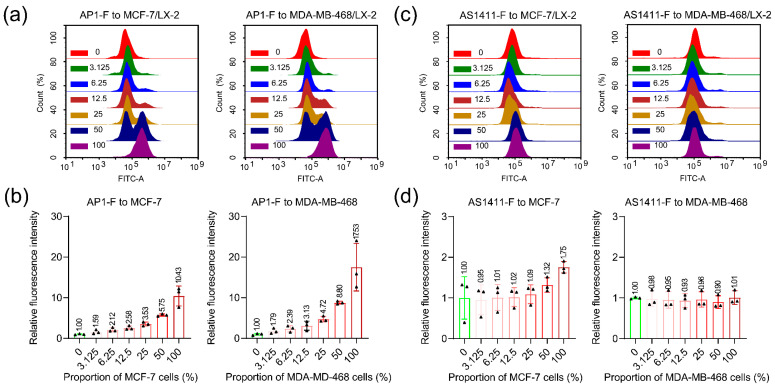
The study examines the effectiveness of AP1-F and AS1411-F in identifying target cells within a dual-cell mixed system (0%, 3.125%, 6.25%, 12.5%, 25%, 50%, 100%). (**a**,**c**) The flow cytometry results of mixed cell populations labeled with AP1-F (1 μM) and AS1411-F (1 μM), respectively. (**b**,**d**) Statistical analysis of fluorescence intensity in mixed cell samples stained with AP1-F and AS1411-F. Detection channel: FITC. The flow cytometry histogram findings were acquired at a constant wavelength of 488 nm/530 nm. Data were presented as the mean ± standard deviation (n = 3).

**Figure 6 cimb-47-00904-f006:**
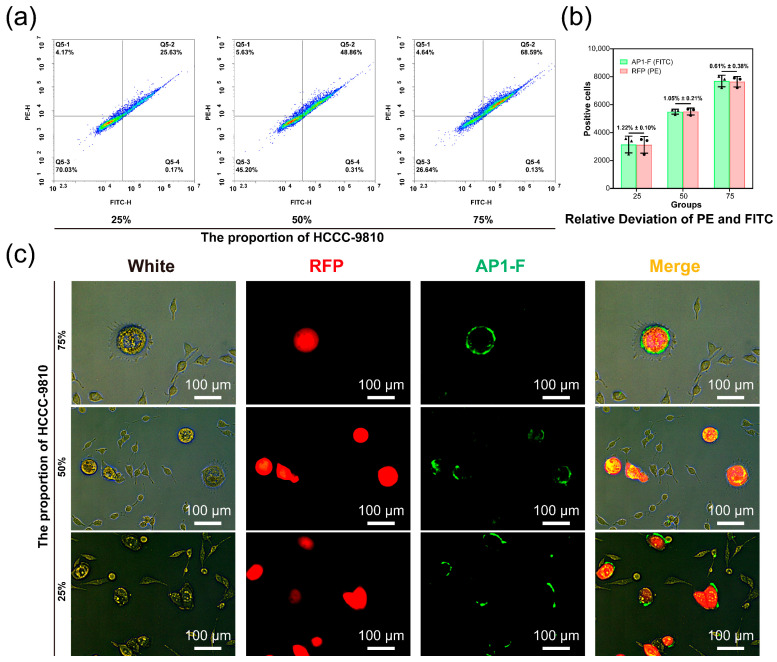
The study involved the analysis of dual-channel flow cytometry for various ratios (25%, 50%, and 75%) of mixed cells tagged with AP1-F and RFP fluorescent proteins. (**a**) The dot plot from flow cytometry displays the double-labeled HCCC-9810. AP1-F detection channel: FITC; RFP fluorescent protein detection channel: PE. (**b**) The figure illustrates the relative deviation in the number of PE- and FITC-labeled positive cells within the same mixed cell sample, using PE as the control. (**c**) The co-localization of AP1-F and RFP fluorescent protein was observed in double-labeled rectal cancer cells HCCC-9810, which were part of mixed cell populations. Scale bar: 100 µm. The histogram results from flow cytometry were obtained using a fixed wavelength of λex/λem = 488 nm/530 nm or λem = 561 nm. The data were presented as the mean ± standard deviation (n = 3).

**Figure 7 cimb-47-00904-f007:**
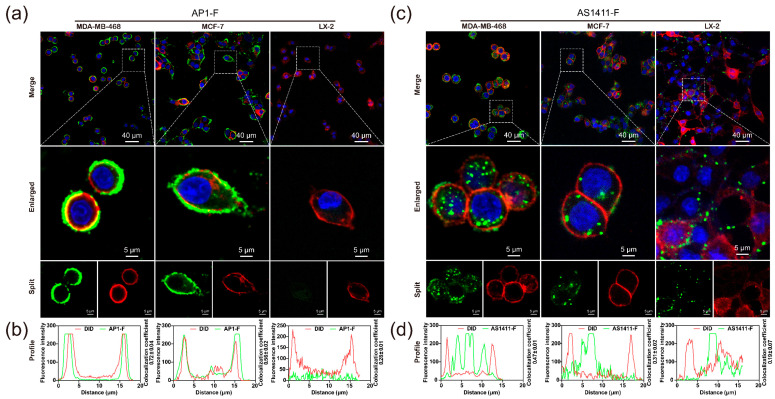
Laser Confocal Imaging demonstrates unique patterns of cellular binding distribution between AP1-F and AS1411-F. (**a**,**c**) Laser Confocal Imaging of breast cancer cells (MCF-7, MDA-MB-468) and normal cells (LX-2) treated with AP1-F or AS1411-F 1 µM. Green channel: λex = 488 nm, λem = 500–530 nm. Blue channel: λex/λem = 405 nm/440–470 nm. Red channel: λex/λem = 561 nm/600–700 nm. Scale bar: 5 µm. (**b**,**d**) Co-localization study of three cellular types Fluorescence profiling. The overlapping confocal imaging was channel-split utilizing ImageJ to acquire independent channels of distinct colors, and Pearson’s Correlation Coefficient Co-localization Analysis was conducted on the fluorescence of the two hues. Data were expressed as the mean ± SD (n = 3).

**Figure 8 cimb-47-00904-f008:**
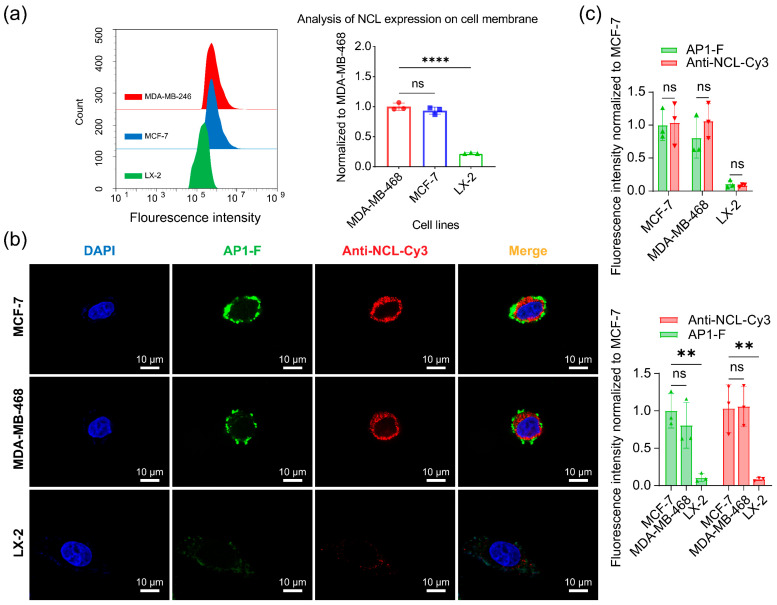
Immunoblotting of cell membranes with co-staining to examine variations in NCL expression across distinct cell membranes. (**a**) Left side: Flow cytometry analysis of NCL expression in three cell membranes. Right side: Statistical graph of NCL expression in cell membranes. (**b**) Dual-visual imaging of NCL expression on the membranes of two breast cancer cells (MCF-7, MDA-MB-468) and a normal cell (LX-2). Green channel: λex/λem = 488 nm/500–530 nm. Blue channel: λex/λem = 405 nm/440–470 nm. Red channel: λex/λem = 561 nm/600–700 nm. Scale bar: 10 µm. (**c**) The statistical analysis was conducted on the fluorescence intensity of AP1-F and anti-NCL-Cy3 double-labeled cancer cells. Data were expressed as the mean ± SD/SEM (n = 3). ** *p* < 0.01, **** *p* < 0.0001 (One-way ANOVA test).

**Figure 9 cimb-47-00904-f009:**
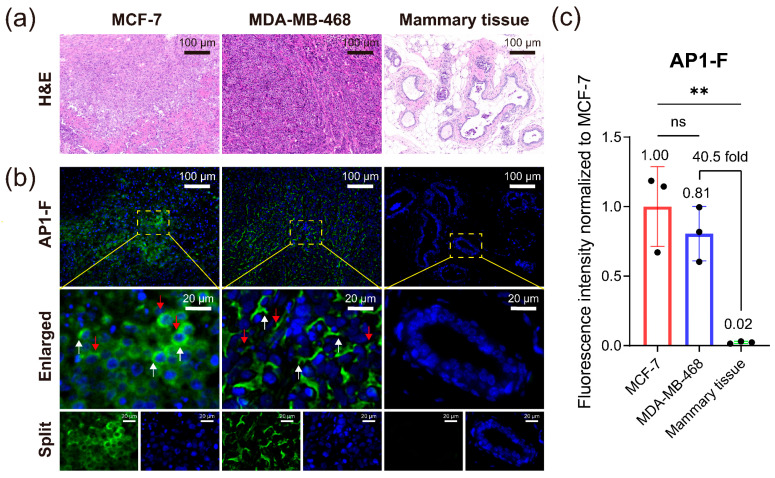
Immunofluorescence staining of pathologic tissue sections for breast cancer. (ER+MCF-7 and triple-negative MDA-MB-468). (**a**) H&E tissue section staining. (**b**) AP1-F staining at 2 µM concentration was performed on two breast cancer pathology tissues and one normal mouse mammary tissue section. White and red arrows correspond to cell membrane pseudopodia and ruffling, respectively. Green channel: λex/λem = 488 nm/500–530 nm. Blue channel: λex/λem = 405 nm/440–470 nm. Scale bar: 20 µm. (**c**) Statistics on the staining fluorescence intensity of different breast cancer tissues and normal mouse tissues using AP1-F. Data were expressed as the mean ± SD (n = 3). ** *p* < 0.01 (One-way ANOVA test).

## Data Availability

The original contributions presented in this study are included in the article and Appendix A. Further inquiries can be directed to the corresponding author.

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
