# Peer review of "Dual-Mode Aptamer AP1-F Achieves Molecular–Morphological Precision in Cancer Diagnostics via Membrane NCL Targeting"

_cimb, 2025, doi:10.3390/cimb47110904_

Round 1

Reviewer 1 Report

Comments and Suggestions for Authors

In this manuscript, Zheng-Lin at al. describe the use of G-quadruplex-forming aptamers previously developed, able to target membrane nucleolin (NCL), for cancer diagnostics. The authors performed several experiments to provide evidence that AP1-F achieves superior specificity compared to AS1411 by avoiding intracellular internalization and maintaining spatial confinement to the cell membrane. This work is interesting and has translational potential. However, a few issues should be clarified or revised before publication.

Major Comments

1. First of all, it is not clear how the aptamer sequences used in this study were initially identified. The rationale for focusing specifically on NCL and the description of the aptamer discovery process must be clarified;

2. The cell line used in section 3.2, i.e. HL-7702, is not reported in the Materials and Methods section and its origin is not properly described. Please check and correct inconsistencies between Methods and Results;

3. In this study, the authors used as “normal” control, ephaptic and liver cells. However, since the system aims to improve intraoperative breast cancer detection, normal breast epithelial cells would be a more relevant control. This limitation should be acknowledged explicitly in the Discussion, since it affects the translational impact;

4. The experiment reported in section 3.4 is insufficiently described. In the Materials and Methods section, it is stated that cells were incubated first with aptamers and then with primary and secondary antibodies. However, in the Results the decrease in fluorescence signal seems to be attributed to aptamer binding. The authors must better describe the assay to avoid
misinterpretations;

5. While the study demonstrates a strong correlation between AP1-F staining and NCL expression, more discussion is needed on possible off-target effects and whether AP1-F interacts with other membrane proteins. Including appropriate controls or references would strengthen the claim of specificity.

Minor Comments

1. Please provide more references to support the background statements on aptamer advantages for diagnostics (Introduction, page 2). Some examples:

  • - Napolitano,E. et al. (2023) Selective light-up of dimeric G-quadruplex forming aptamers for efficient VEGF165 detection. J. Biol. Macromol., 224, 344–357. https://doi.org/10.1016/j.ijbiomac.2022.10.128
  • Xin Z. et al. (2021) A Fluorescent Aptasensor Based on Assembled G-Quadruplex and Thioflavin T for the Detection of Biomarker VEGF165, 9, 764123.

https://doi.org/10.3389/fbioe.2021.764123

  • Zamay G., Visualization of Brain Tumors with Infrared-Labeled Aptamers for Fluorescence-Guided Surgery, 146, 24989-25004.

https://doi.org/10.1021/jacs.4c06716

2. The figures are informative, but some legends are unclear. For example, Figure 4 should specify some details of the assay;

3. There is a typo in the author names.

Author Response

Comments 1: [1. First of all, it is not clear how the aptamer sequences used in this study were initially identified. The rationale for focusing specifically on NCL and the description of the aptamer discovery process must be clarified]

Response 1: Thank you for pointing this out. We agree with this comment. Therefore, we have described the inspiration behind the design of the aptamers used in this study and their prior research foundation, detailing the results of their discovery and structural validation, along with supplementary literature from previous studies for your review. [Studies demonstrate that G-end telomeric RNA sequences can generate G4 structures, with loop length serving as a critical determinant affecting G4 affinity for NCL [30]. Utilizing nuclear magnetic resonance, circular dichroism, and gel electrophoresis techniques, we discerned that the human telomeric RNA sequence r(GUUAGGGU) establishes a distinctive topological structure focused on a very stable G-quadruplex. Consequently, we engineered RNA sequences with differing loop lengths and previ-ously established that these aptamers assume distinct G4 conformations and display a strong binding affinity for NCL (Table S1) [31, 32].  Page: 6, Paragraph: 3.1, line: 235-242].

Comments 2: [2. The cell line used in section 3.2, i.e. HL-7702, is not reported in the Materials and Methods section and its origin is not properly described. Please check and correct inconsistencies between Methods and Results;]

Response 2: Thank you for pointing this out. We agree with this comment. We have supplemented the Methods and Materials section with information on the “HL-7702” cells and verified their sourcing. The modified sections are highlighted in red for your reference. Page: 2, Paragraph: 2.1, line: 86.

Comments 3: [3. In this study, the authors used as “normal” control, ephaptic and liver cells. However, since the system aims to improve intraoperative breast cancer detection, normal breast epithelial cells would be a more relevant control. This limitation should be acknowledged explicitly in the Discussion, since it affects the translational impact;]

Response 3: Agree. Therefore, we have added a detailed discussion of this issue in the final paragraph of the discussion section, and we cannot emphasize this point enough. [This study design possesses specific constraints, chiefly stemming from the challenges associated with acquiring cancer cells and normal cells from the same tissue source, as well as clinical cancer tissue samples for detection in mixed specimens. This study employed xenograft models as a replacement. Ex vivo cancer cells inhabit a dis-tinct internal environment relative to in situ breast cancer cells in vivo, which may re-sult in biological variations that could, to some degree, influence the translational relevance of the findings page 17, paragraph 5, and line: 597-602].

Comments 4: [4. The experiment reported in section 3.4 is insufficiently described. In the Materials and Methods section, it is stated that cells were incubated first with aptamers and then with primary and secondary antibodies. However, in the Results the decrease in fluorescence signal seems to be attributed to aptamer binding. The authors must better describe the assay to avoid
misinterpretations;]

Response 4: Thank you for pointing this out. We agree with this comment. Therefore, we have added the specific steps and references for this experiment in Section 2.6 of the Methods, and revised the presentation of the corresponding results to more clearly convey the experimental findings and conclusions. [Employing a flow cytometry-based competitive binding displacement assay, as documented in previous studies, we examined whether APs aptamers inhibit the binding of specific antibodies to NCL similarly to the established NCL-binding aptamer (AS1411), thereby confirming their specific recognition of cancer cell binding targets [28].  Page: 4, paragraph: 2.6, line: 147-151; Establish three antibody concentration gradients (1 μg, 0.5 μg, and 0.25 μg). For each antibody concentration, create five groups: Blank control group (Antibodies only), Negative control group (Mutant: 4 μM), Treatment group (AP1: 2 μM, 4 μM), Positive control group (AS1411: 4 μM). After incubating cancer cells with relevant samples from various groups, proceed to incubate them with NCL antibody, followed by flow cy-tometry analysis. Cancer cells from various therapy groups were subsequently treated with the NCL primary antibody, followed by the fluorescent secondary antibody. The results demonstrate that the fluorescence intensity of both cancer cell lines diminished in a dose-dependent fashion with escalating AP1 concentration. At an AP1 concentra-tion of 4 μM, fluorescence intensity diminished by nearly 50% compared to the PBS control group (p<0.001), analogous to the inhibitory impact reported in the positive control group (AS1411). This indicates that AP1 effectively blocks subsequent antibody binding to NCL through specific binding to NCL. Conversely, cancer cells in the nega-tive control group (Mutant) exhibited no substantial decrease in fluorescence intensity, signifying their absence of particular binding affinity to NCL. These additional inves-tigations further corroborate its potential significance as an essential instrument for detecting NCL on cell surfaces and emphasize the critical function of the G4 structure in identifying cancer cells. In overall, both MCF-7 and MDA-MB-468 cells demonstrated a marked decrease in fluorescence intensity at varying antibody doses post-treatment with AP1 and AS1411, signifying that both drugs successfully inhibited further anti-body binding after associating with NCL (Figure 4). This outcome validates AP1's ex-clusive affinity for NCL over other targets at the cellular level and demonstrates that both breast cancer cell lines have analogous competitive response patterns to AP1-NCL binding. - Page: 9, paragraph: 3.4, line: 322-344].

Comments 5: [5. While the study demonstrates a strong correlation between AP1-F staining and NCL expression, more discussion is needed on possible off-target effects and whether AP1-F interacts with other membrane proteins. Including appropriate controls or references would strengthen the claim of specificity.]

Response 5: Thank you for pointing this out. We agree with this comment. To this end, we conducted an in-depth discussion in the discussion section and cited relevant literature to support this viewpoint. [Moreover, whereas competitive binding studies reveal a substantial association between AP1-F labeling and NCL expression, additional research is necessary to examine potential off-target effects and to determine if AP1-F interacts with other membrane proteins [48]. - Page: 16, paragraph: 5, line: 603-605].

Minor comments

Comments 1: [1. Please provide more references to support the background statements on aptamer advantages for diagnostics (Introduction, page 2). Some examples: Napolitano,E. et al. (2023) Selective light-up of dimeric G-quadruplex forming aptamers for efficient VEGF165 detection. J. Biol. Macromol., 224, 344–357. https://doi.org/10.1016/j.ijbiomac.2022.10.128

Xin Z. et al. (2021) A Fluorescent Aptasensor Based on Assembled G-Quadruplex and Thioflavin T for the Detection of Biomarker VEGF165, 9, 764123. https://doi.org/10.3389/fbioe.2021.764123

Zamay G., Visualization of Brain Tumors with Infrared-Labeled Aptamers for Fluorescence-Guided Surgery, 146, 24989-25004. https://doi.org/10.1021/jacs.4c06716]

Response 1: Agree. Following discussion and analysis, we have added three references to the second paragraph of the Introduction section highlighting the advantages of natural aptamer properties. The cited references are now highlighted in red within the article's reference list. Page: 2; 19, paragraph: 2, line: 57; 701-707.

Comments 2: [2. The figures are informative, but some legends are unclear. For example, Figure 4 should specify some details of the assay;]

Response 2: Thank you for pointing this out. We agree with this comment. We have provided more detailed annotations for the images in the article. The revised sections are highlighted in red for your review. [Primary antibody concentration (1 μg/100 μL, 0.5 μg/100 μL, and 0.25 μg/100 μL). Alexa Fluor 488-conjugated anti-rabbit IgG secondary antibody (5 µg/mL). Groups: Blank control group (Antibodies only), Negative control group (Mutant: 4 μM), Treatment group (AP1: 2 μM, 4 μM), Positive control group (AS1411: 4 μM).]- Page: 9, paragraph: 3.4, line: 247-351.

Comments 3: [3. There is a typo in the author names.]

Response 3: Agree. We sincerely appreciate you bringing this issue to our attention. We have now corrected and verified the author's name in this article. - Page: 1, paragraph: 2, line: 4.

4. Response to Comments on the Quality of English Language

Point 1: The English is fine and does not require any improvement.

Response 1: Sincerely thank you for your review.

Reviewer 2 Report

Comments and Suggestions for Authors

In the paper, “Dual-Mode Aptamer AP1-F Achieves Molecular-Morphological Precision in Cancer Diagnostics via Membrane NCL Targeting,” the authors have designed and tested a novel aptamer targeting membrane localized nucleolin protein. The novel aptamer, AP1-F, contains a fluorescent tag and a repetitive guanine-rich sequence which folds into a G-quadruplex structure. AP1-F binds to membrane-localized nucleolin protein, which is a known G-quadruplex structure binding protein, and provides a handle for detection of cells overexpressing nucleolin protein. Since various cancer cells overexpress nucleolin, it is a great tool for cancer cell detection and targeting.

The manuscript is well-structured, and the experimental data are clean and support the authors’ claim. The authors have performed all the necessary experiments required to validate their hypothesis. The AP1-F provides significant specificity over the previously identified G-quadruplex aptamer AS1411. Unlike AS1411, AP1-F does not penetrate cancer cells and remains surface-bound, which is advantageous for cancer cell detection and targeting without eliciting non-specific side effects, as was found in the case of AS1411 clinical trials. The authors convincingly showed that AP1-F specifically targets nucleolin on the plasma membrane and detects cancer cells with high specificity compared to non-cancerous cells. The result and discussion sections are nicely written and explains the experimental data very well. The authors proactively stated several limitations of the study and future directions. The cited references are relevant and up to date. I do not have any comments or suggestions for the authors."

Author Response

Response to Reviewer 2 Comments

1. Summary

Thank you very much for taking the time to review this manuscript.

2. Questions for General Evaluation

Reviewer’s Evaluation

Response and Revisions

Does the introduction provide sufficient background and include all relevant references?

Yes

Are all the cited references relevant to the research?

Yes

Is the research design appropriate?

Yes

Are the methods adequately described?

Yes

Are the results clearly presented?

Yes

Are the conclusions supported by the results?

Yes

3. Response to Comments on the Quality of English Language

Point 1:

Response 1:  Thank you for your constructive suggestions. We will carefully consider your feedback and refine the wording to ensure a more professional presentation.

Reviewer 3 Report

Comments and Suggestions for Authors

In this study, the authors used a fluorescent AP1-F, a G-quadruplex (G4)-structured aptamer, for specific quantification of membrane NCL on live cancer cells. They performed several experiments to demonstrate its utility in cancer diagnostics. This research are important and promising; however, I found some mistakes in this manuscript.

  1. The main objection concerns incorrectly conducted MST experiments and, therefore, incorrect conclusions. The correct Kd constant is determined only for AP1 and AP2: on the binding curves, two flat regions can be observed (one for the unbound state and the second for the bound state). Although the MST instrument is also able to determine Kd for the other aptamers, even if other parameters such as Kd confidence, response amplitude, etc.. (Table S2) are within the required range, these measurements are not correct because of the lack of a second flat region – thie bound state. Higher concentrations of ligand (NCL) than 2 µM should be used – then the binding curves should also demonstrate two flat regions.

This is especially unfortunate because the AP3 aptamer was selected for some further studies due to its highest affinity, as the authors emphasize. Moreover, the conclusions of the experiment presented in part 3.4. Confirm Specificity of Target Recognition, state:

“NCL protein in vivo tests may impact its binding capacity with ligands due to conformational changes and spatial hindrance effects at the binding site. This may be the reason why AP1 operates better than AP3 in selective applications at the cellular level, even if AP3 showed the strongest NCL affinity in MST tests.”- This is not true if the Kd of binding is incorrectly determined. Therefore, I recommend performing MST experiments with higher concentrations of NCL.

  1. Figure 6a –“ Dual-channel flow cytometry analysis of different ratios of mixed cells labeled with AP1-F and RFP fluorescent proteins” – The presentation of this experiment is incorrect. This flow cytometry experiment should be presented as a dot plot with AP1-F fluorescence on one axis (e.g., X axis) and RFP protein fluorescence at second one (e.g., Y axis). Then, we could precisely observe single- and double- labeled cells. In the current visualization, we cannot determine whether there are double- labeled cells.
  2. Figure 8a – Please provide the count (Y axis) for each cell line separately and in the proper scale (as shown on the adjacent chart).

Author Response

For research article

Response to Reviewer 3 Comments

1. Summary

Thank you very much for taking the time to review this manuscript.

2. Questions for General Evaluation

Reviewer’s Evaluation

Response and Revisions

Does the introduction provide sufficient background and include all relevant references?

Yes

Are all the cited references relevant to the research?

Can be improved

Response

Is the research design appropriate?

Can be improved

Response

Are the methods adequately described?

Can be improved

Revisions

Are the results clearly presented?

Can be improved

Revisions

Are the conclusions supported by the results?

Yes

3. Point-by-point response to Comments and Suggestions for Authers

Major comments

Comments 1: [1. The main objection concerns incorrectly conducted MST experiments and, therefore, incorrect conclusions. The correct Kd constant is determined only for AP1 and AP2: on the binding curves, two flat regions can be observed (one for the unbound state and the second for the bound state). Although the MST instrument is also able to determine Kd for the other aptamers, even if other parameters such as Kd confidence, response amplitude, etc. (Table S2) are within the required range, these measurements are not correct because of the lack of a second flat region – thie bound state. Higher concentrations of ligand (NCL) than 2 µM should be used – then the binding curves should also demonstrate two flat regions.

This is especially unfortunate because the AP3 aptamer was selected for some further studies due to its highest affinity, as the authors emphasize. Moreover, the conclusions of the experiment presented in part 3.4. Confirm Specificity of Target Recognition, state:

“NCL protein in vivo tests may impact its binding capacity with ligands due to conformational changes and spatial hindrance effects at the binding site. This may be the reason why AP1 operates better than AP3 in selective applications at the cellular level, even if AP3 showed the strongest NCL affinity in MST tests.”- This is not true if the Kd of binding is incorrectly determined. Therefore, I recommend performing MST experiments with higher concentrations of NCL.]

Response 1:Agree. Thank you for your suggestion. To ensure a more complete MST curve and enhance the accuracy of the measurement results, we conducted binding affinity tests at higher concentrations (up to 4μM). The updated results have been incorporated into the manuscript and highlighted in red for your review..Page: 6, paragraph: 3.1, lines: 252-253]

Comments 2: [2. Figure 6a –“ Dual-channel flow cytometry analysis of different ratios of mixed cells labeled with AP1-F and RFP fluorescent proteins” – The presentation of this experiment is incorrect. This flow cytometry experiment should be presented as a dot plot with AP1-F fluorescence on one axis (e.g., X axis) and RFP protein fluorescence at second one (e.g., Y axis). Then, we could precisely observe single- and double- labeled cells. In the current visualization, we cannot determine whether there are double- labeled cells.]

Response 2: Thank you for pointing this out. We agree with this comment. To present the results more clearly and intuitively, we have replaced the original flow cytometry histogram with a scatter plot. The modified content has been annotated in the text for your review. [Double-labeled HCCC-9810 flow cytometry dot plot. AP1-F detection channel: FITC; RFP fluorescent protein detection channel: PE. Figure 6a; Page: 11, paragraph: 3.6, line: 402-403]

Comments 3: [3. Figure 8a – Please provide the count (Y axis) for each cell line separately and in the proper scale (as shown on the adjacent chart).]

Response 3: Thank you for your suggestion. [We have adjusted the coordinates of the modified image. The purpose of this experiment is to compare the numerical values of the average fluorescence intensity across different cell lines (i.e., the values on the X-axis corresponding to each cell line). To make the comparison more intuitive, we normalized the Y-axis (Count) and combined them onto a single Y-axis. Page: 14, paragraph: 3.8, line: 479]

Round 2

Reviewer 1 Report

Comments and Suggestions for Authors

The authors have provided clear responses to all the questions and have integrated the text accordingly.

There is a typo in the caption of Figure 5. "AS1411-F (μM)" should be "AS1411-F (1μM)".

Author Response

For research article

Response to Reviewer 1 Comments

1. Summary

Thank you very much for taking the time to review this manuscript once again. We have corrected the spelling errors you pointed out, and the corresponding revisions have been marked in the resubmitted document.

2. Point-by-point response to Comments and Suggestions for Authers

Major comments

Comments 1: [1. There is a typo in the caption of Figure 5. "AS1411-F (μM)" should be "AS1411-F (1μM)".]

Response 1: We sincerely appreciate you pointing out the spelling error. We have made the necessary correction and highlighted it in red for your review.

Reviewer 3 Report

Comments and Suggestions for Authors

The authors have addressed the issues I raised previously.

However additional minor editor revision should be done:

  • The name of the method of solid-phase chemical synthesis of aptamers is wrong;

Line 97 – “standard phosphonamidite solid-phase chemistry on an automated DNA/RNA synthesizer”

It should be “phosphoramidite” or “H-phosphonate”, depending on which one was used.

  • Line 358: “at varying ratios,…” – please provide these ratios
  • Line 383: ” at a specific ratio, …” Please provide this ratio.
  • Figure 6 and the caption for Figure 6: Please provide the values for the “ different ratios of mixed cells” and add appropriate descriptions to the graphs in (a)
  • Please check carefully all the captions for the figures and add the values concentrations/ratios for used compound in those places where they are missing.
  • Language should be improved, as there are many grammar errors, e.g.

Line 44-47: “While conventional multi-antibody workflows (e.g., ER/PR/HER2 panels) suffer from three inherent limitations: time-intensive sequential processing, cumulative analytical variability (>30% inter-batch coefficient variation), and architectural disruption from membrane permeabilization”

Line 197: “Digest MCF-7 and LX-2 cells in the logarithmic growth phase with 0.25% trypsin,...”

Line 309: “NCL protein in vivo tests may impact its binding capacity with ligands due to conformational changes and spatial hindrance effects at the binding site.”

Lines 315-317: “To validate the cell-level specificity of aptamer APs targeting NCL and assess their targeting adaptability, this study selected two breast cancer cell lines with different molecular subtypes (MCF-7 and MDA-MB-468) and conducted competitive binding experiments using flow cytometry combined with immunofluorescence technology.”

Lines 323-328: “Establish three antibody concentration gradients (1 μg, 0.5 μg, 0.25 µg). For each antibody concentration, create five groups: Blank control group (Antibodies only), Negative control group (Mutant: 4 μM), Treatment group (AP1: 2 μM, 4 μM), Positive control group (AS1411: 4 μM). After incubating cancer cells with relevant samples from various groups, proceed to incubate them with NCL antibody, followed by flow cytometry analysis.”

And many others…… , such as “We” with capital letter in the middle of a sentence (line 383).

Therefore, I recommend language editing of the entire manuscript by an expert professional prior to final acceptance.

Comments on the Quality of English Language

There are many grammar errors, e.g.

Line 44-47: “While conventional multi-antibody workflows (e.g., ER/PR/HER2 panels) suffer from three inherent limitations: time-intensive sequential processing, cumulative analytical variability (>30% inter-batch coefficient variation), and architectural disruption from membrane permeabilization”

Line 197: “Digest MCF-7 and LX-2 cells in the logarithmic growth phase with 0.25% trypsin,...”

Line 309: “NCL protein in vivo tests may impact its binding capacity with ligands due to conformational changes and spatial hindrance effects at the binding site.”

Lines 315-317: “To validate the cell-level specificity of aptamer APs targeting NCL and assess their targeting adaptability, this study selected two breast cancer cell lines with different molecular subtypes (MCF-7 and MDA-MB-468) and conducted competitive binding experiments using flow cytometry combined with immunofluorescence technology.”

Lines 323-328: “Establish three antibody concentration gradients (1 μg, 0.5 μg, 0.25 µg). For each antibody concentration, create five groups: Blank control group (Antibodies only), Negative control group (Mutant: 4 μM), Treatment group (AP1: 2 μM, 4 μM), Positive control group (AS1411: 4 μM). After incubating cancer cells with relevant samples from various groups, proceed to incubate them with NCL antibody, followed by flow cytometry analysis.”

And many others…… , such as “We” with capital letter in the middle of a sentence (line 383).

Therefore, I recommend language editing of the entire manuscript by an expert professional prior to final acceptance.

Author Response

For research article

Response to Reviewer 3 (Round 2) Comments

1. Summary

Thank you very much for taking the time to review this manuscript.

2. Point-by-point response to Comments and Suggestions for Authers

Major comments

Comments 1: [1. It should be “phosphoramidite” or “H-phosphonate”, depending on which one was used.]

Response 1: Thank you for pointing this out. We agree with this comment. [We have revised the phrase and highlighted it in red for your review. Page: 3, paragraph: 2.2, lines:99]

Comments 2: [2. Line 358: “at varying ratios,…” – please provide these ratios.]

Response 2: Thank you for pointing this out. We agree with this comment. [We have added the specific percentage values for that location. The modified content has been highlighted in red for your review. Page: 9, paragraph: 3.5, line: 350]

Comments 3: [3. Line 383: ” at a specific ratio, …” Please provide this ratio.]

Response 3: Thank you for pointing this out. We agree with this comment. [We have added the specific percentage values for that location. The modified content has been highlighted in red for your review. Page: 10, paragraph: 3.6, line: 375]

Comments 4: [4. Figure 6 and the caption for Figure 6: Please provide the values for the “ different ratios of mixed cells” and add appropriate descriptions to the graphs in (a)

Please check carefully all the captions for the figures and add the values concentrations/ratios for used compound in those places where they are missing.]

Response 4: Thank you for pointing this out. We agree with this comment. [We have added the corresponding scales to the images and captions, with the additions highlighted in red for your review. Additionally, we have reviewed and carefully revised all captions, with the modified captions highlighted in red for your review. Figure 6, paragraph: 3.6, line:393]

Comments 5: [5. Language should be improved, as there are many grammar errors, e.g.

Line 44-47: “While conventional multi-antibody workflows (e.g., ER/PR/HER2 panels) suffer from three inherent limitations: time-intensive sequential processing, cumulative analytical variability (>30% inter-batch coefficient variation), and architectural disruption from membrane permeabilization”.]

Response 5: Thank you for pointing this out. We agree with this comment. [We have revised the sentence for grammatical accuracy and enhanced its professional expression. The revised portions are highlighted in red for your review. Page: 2, paragraph: 1, line:45-48]

Comments 6: [6. Line 197: “Digest MCF-7 and LX-2 cells in the logarithmic growth phase with 0.25% trypsin,...”.]

Response 6: Thank you for pointing this out. We agree with this comment. [We have revised the sentence for grammatical accuracy and enhanced its professional expression. The revised portions are highlighted in red for your review. Page: 5, paragraph: 2.9, line:200-202]

Comments 7: [7. Line 309: “NCL protein in vivo tests may impact its binding capacity with ligands due to conformational changes and spatial hindrance effects at the binding site.”]

Response 7: Thank you for pointing this out. We agree with this comment. [We have revised the sentence for grammatical accuracy and enhanced its professional expression. The revised portions are highlighted in red for your review. Page: 8, paragraph: 3.4, line:313-315]

Comments 8: [8. Lines 315-317: “To validate the cell-level specificity of aptamer APs targeting NCL and assess their targeting adaptability, this study selected two breast cancer cell lines with different molecular subtypes (MCF-7 and MDA-MB-468) and conducted competitive binding experiments using flow cytometry combined with immunofluorescence technology.”]

Response 8: Thank you for pointing this out. We agree with this comment. [We have revised the sentence for grammatical accuracy and enhanced its professional expression. The revised portions are highlighted in red for your review. Page: 8, paragraph: 3.4, line:317-320]

Comments 9: [9. Lines 323-328: “Establish three antibody concentration gradients (1 μg, 0.5 μg, 0.25 µg). For each antibody concentration, create five groups: Blank control group (Antibodies only), Negative control group (Mutant: 4 μM), Treatment group (AP1: 2 μM, 4 μM), Positive control group (AS1411: 4 μM). After incubating cancer cells with relevant samples from various groups, proceed to incubate them with NCL antibody, followed by flow cytometry analysis.”]

Response 9: Thank you for pointing this out. We agree with this comment. [We have revised the sentence for grammatical accuracy and enhanced its professional expression. The revised portions are highlighted in red for your review. Page: 9, paragraph: 3.4, line:321-324]

Comments 10: [10. And many others…… , such as “We” with capital letter in the middle of a sentence (line 383).

Therefore, I recommend language editing of the entire manuscript by an expert professional prior to final acceptance.”]

Response 10: Thank you for pointing this out. We agree with this comment. [To correct grammatical errors and inappropriate wording throughout the text, we have conducted a comprehensive review of the syntax and expression style. Paragraphs containing imprecise or non-professional language have been systematically optimized. Relevant revisions are highlighted in red for your reference. Page: 10, paragraph: 3.6, line:376]

We are grateful for your guidance and believe these revisions address all concerns. For further clarification, please contact the corresponding authors:

Prof. Chao-da Xiao: xcd@gmc.edu.cn

Prof. Xiang-chun Shen: sxc@gmc.edu.cn

Sincerely,

Chao-da Xiao and Xiang-chun Shen

Guizhou Medical University

October 24, 2025